# The Incremental Role of Multiorgan Point-of-Care Ultrasounds in the Emergency Setting

**DOI:** 10.3390/ijerph20032088

**Published:** 2023-01-23

**Authors:** Antonello D’Andrea, Carmen Del Giudice, Dario Fabiani, Adriano Caputo, Francesco Sabatella, Luigi Cante, Stefano Palermi, Alfonso Desiderio, Ercole Tagliamonte, Biagio Liccardo, Vincenzo Russo

**Affiliations:** 1Department of Cardiology and Intensive Coronary Care, Umberto I Hospital, 84014 Nocera Inferiore, Italy; 2Division of Cardiology, Department of Traslational Medical Sciences, University of Campania Luigi Vanvitelli, 80131 Naples, Italy; 3Public Health Department, University of Naples Federico II, 80131 Naples, Italy

**Keywords:** POCUS, point-of-care, emergency medicine (EM), echocardiography, lung ultrasound (LUS), VE x US, congestion, deep-vein thrombosis (DTV)

## Abstract

Point-of-care ultrasonography (POCUS) represents a goal-directed ultrasound examination performed by clinicians directly involved in patient healthcare. POCUS has been widely used in emergency departments, where US exams allow physicians to make quick diagnoses and to recognize early life-threatening conditions which require prompt interventions. Although initially meant for the real-time evaluation of cardiovascular and respiratory pathologies, its use has been extended to a wide range of clinical applications, such as screening for deep-vein thrombosis and trauma, abdominal ultrasonography of the right upper quadrant and appendix, and guidance for invasive procedures. Moreover, recently, bedside ultrasounds have been used to evaluate the fluid balance and to guide decongestive therapy in acutely decompensated heart failure. The aim of the present review was to discuss the most common applications of POCUS in the emergency setting.

## 1. Introduction

In 2001, the American College of Emergency Physicians (ACEP) published the first Emergency Ultrasound Guidelines with the aim to describe the principal point-of-care ultrasonography (POCUS) applications and to underline the importance of continuing education and training in the use of emergency ultrasounds. These guidelines then expanded together with the growing use of focused ultrasounds [1]. POCUS represents an important diagnostic tool which is performed at the bedside, thus allowing physicians to rapidly evaluate patients admitted into the emergency department (ED) and to make clinical decisions [2]. POCUS is used for goal-directed echocardiography, lung ultrasounds, screening for deep-vein thrombosis in people suspected to have a pulmonary embolism, abdominal ultrasonography, intracranial pressure and trauma monitoring, and guidance for invasive procedures [3,4]. The aim of this review was to summarize the most common applications of POCUS in the emergency setting.

## 2. Lung Ultrasounds in the Emergency Department

### 2.1. Technical Equipment and Methodology

Lung ultrasound (LUS) examinations can be performed with the use of probes of different frequencies based on the lung region to be explored. A high-frequency (7.5–10 MHz) linear probe, thanks to it having a greater spatial resolution than its depth of penetration, is useful for performing a detailed exam of the chest wall and pleurae. Conversely, a low-frequency (3.5–5 MHz) curvilinear probe has a greater depth of penetration and is suitable for examining the parenchymal structures below the pleurae. Lichtenstein et al. [5] recommended the use of a 5 MHz microconvex probe for bedside LUSs. Many studies suggest the use of a single probe for a complete LUS examination in the emergency and critical care setting [6]. The probes are placed over the intercostal spaces along the longitudinal plane, perpendicular to the ribs [7,8], offering an acoustic window into the lung parenchyma. The 2012 and 2022 international consensuses have proposed an eight-region model for LUS examinations in the emergency department [6,8]. Based on this protocol, each hemithorax is divided into two anterior and lateral regions by the parasternal line and the anterior and posterior axillary lines. Each of these regions is subdivided into upper and basal segments, leading to a total of eight regions. Alternatively, Lichtenstein et al. proposed the Bedside Lung Ultrasound in Emergency (BLUE) protocol for LUS examinations of patients presenting to the emergency department with dyspnea [5]. The BLUE protocol identifies three points of interest per hemithorax to be examined, or so-called BLUE points [5], and quickly provides three clinical diagnostic profiles. LUS examinations include two main modalities: B-mode (brightness mode), which generates two-dimensional images, and M-mode, which is useful for examining the motion of the lung surfaces towards and away from the probe. Doppler techniques are not usually used in a standard exam. Moreover, LUS examinations are based on two types of images: artifacts and real images. Artifacts (A-lines and B-lines) could be normal or pathological. They are due to the higher acoustic impedance differences between the air and the superficial lung tissues and are generated by the pleural surface [9]. Conversely, real images (such as pleural effusion and lung consolidations) are always pathological signs.

### 2.2. Pleural Disorders

#### 2.2.1. Pneumothorax

LUS examinations allow for the detection of trapped air within the pleural space. Three signs are suggestive of a pneumothorax (PNX) diagnosis.

**(A)** Abolition of lung sliding

Lung sliding is represented by the physiological movement of the parietal pleura against the visceral pleura during the respiratory cycle, which is visualized in B-mode images [10]. The abolition of lung sliding suggests a PNX with a high negative predictive value (sensitivity of 95% and negative predictive value of 100%) in the general population [11], while its presence can accurately rule out a PNX diagnosis (sensitivity of 95%, specificity of 91%, and negative predictive value of 100%) [11]. Atelectasis, one-lung intubation, acute respiratory distress syndrome (ARDS), pneumonia, pulmonary fibrosis, and cardiopulmonary arrest, which reduce regional lung ventilation, can cause false-positive diagnoses. For this reason, the positive predictive value of lung sliding abolition for PNX diagnoses is reduced to 56% in critically ill patients [12] and to 27% in the case of acute respiratory failure [5]. Hence, the abolition of lung sliding is not always suggestive of a PNX, and this sign, when isolated, is not sufficient for PNX diagnoses.

**(B)** A-lines

A-lines are horizontal hyperechoic lines localized below the pleural line and repeated at equal intervals (Figure 1). They represent the reverberation artifacts arising from the pleurae and correspond to normal lung ventilation [13]. A-lines are always present in PNXs (sensitivity of 100%) [14] but are not specific because they are also generated by physiological aeration or other pathological conditions (such as chronic obstructive pulmonary disease (COPD) exacerbation and obstructed airways). An A-line presence with the associated abolition of lung sliding is highly suggestive of a PNX in trauma patients (sensitivity of 98% and specificity of 99%) [15].

**(C)** Lung point

In the M-mode view, the “seashore sign” is defined as the presence of straight lines above the pleural line with an associated granular patter below it. This sign confirms lung ventilation and sliding [5] and could be helpful in the critical care setting, where sliding is reduced or absent [16]. Conversely, the absence of pleural movement during respiratory cycles causes straight lines both above and below the pleural line, a pattern defined as the “stratosphere sign”, in the M-mode view. The lung point is defined as the point where the normal lung (“seashore sign”) replaces the PNX air trapped (“stratosphere sign”) during deep inspiration (Figure 2) [17]. Its position must be searched for with a probe in the thorax. This sign is specific for PNX diagnoses (sensitivity of 66% and specificity of 100%) [12] and allows a semiquantitative evaluation of PNX extension (>30% if localized below the midaxillary line) [13].

#### 2.2.2. Pleural Effusion

Pleural effusion is defined as a hypoechoic/anechoic structure without the presence of air inside which presents during expiration and inspiration (Figure 3) [18]. The probe is placed on the posterior region behind the posterior axillary line of a supine patient. In the critical care setting, when it is not always possible to distinguish the pathognomonic hypoechoic/anechoic structure (such as in case of a hemothorax or pyothorax), it could be useful to look for the “sinusoid sign”, which consists in the movement of the lung towards the pleural line during inspiration and away from the pleural line during expiration [19]. LUSs allow for the detection of pleural effusion with a high diagnostic accuracy (sensitivity of 93% and specificity of 97%) [20]; in addition, the “sinusoid sign” presents a high specificity for pleural effusion diagnoses (specificity of 97%) [21]. Various studies have supported that LUSs have a better diagnostic accuracy than chest X-rays (CXR) for evaluating pleural effusion [6,22]. When pleural effusion is abundant, the adjacent lung region is seen to be consolidated and floating within the pleural space [23]. In the presence of pleural effusion localized in the basal lung regions, it is possible to identify, as anatomical landmarks, the spleen, liver, and diaphragm. A color Doppler showing the intrasplenic and intrahepatic blood vessels is a useful tool for distinguishing the spleen and liver tissues from pleural effusion. LUS examinations also allow for the quantification of pleural effusion: a distance between the lung and posterior chest wall, measured at the lung basal regions during end-expiration or end-inspiration with the patient in a supine position, of ≥50 mm is highly predictive of a pleural effusion volume of ≥500 mL [24,25]. There is a high correlation degree between the pleural effusion volume assessed through computed tomography (CT) and that which was assessed through an LUS exam [26]. Finally, LUS exams allow for guidance during thoracentesis procedures at the bedside with a high success rate [27].

### 2.3. Lung Parenchymal Disorders

#### 2.3.1. Interstitial Syndrome

B-lines are vertical hyperechoic lines derived from the pleural line (Figure 4), obliterating physiological A-lines and moving synchronously with lung sliding until the edge of the screen [6,19]. They are artifacts that appear due to the interaction of the ultrasound with the air–fluid interface of the lung tissue, and their presence correlates with thickened subpleural interlobular septa [14]. A maximum of two B-lines per scan can be present in physiological conditions [9]. The presence of B-lines allows one to rule out a PNX diagnosis with a high negative predictive value (sensitivity of 100%, specificity of 60%, and negative predictive value of 100%) [14]. An increased number of B-lines is associated with an impaired air–tissue ratio of the examined lung area and with a strong correlation with the tissue density measured through CT [13]. The presence of three or more B-lines per scan without A-lines is always pathological and indicates a specific pattern called a “B-pattern” [28]. This ultrasonographic pattern is suggestive of an interstitial syndrome diagnosis (93% accuracy compared with CXR and 100% compared with CT) [9], also allowing differential diagnoses of COPD exacerbation (sensitivity of 100% and specificity of 92%) [25,29]. The number of B-lines per scan correlates with the degree of lung aeration loss [9,20]. The distribution of lung B-lines allows for the evaluation of the differential diagnoses of various clinical conditions [8,30]. The presence of a diffuse B-pattern (at least two regions per hemithorax) with a homogeneous and gravitational (basal-to-apex gradient) distribution, regular thin pleurae, and normal sliding indicates cardiogenic edema [31]. Conversely, a diffuse B-pattern with a nonhomogeneous distribution (surfaces of normal lung parenchyma spared), irregular and thickened pleurae, anterior subpleural consolidations, and absent or reduced lung sliding points to ARDS [31].

#### 2.3.2. Lung Consolidation

Lung consolidation corresponds to a complete loss of lung tissue aeration, which favors ultrasound transmission and is a common finding in various clinical conditions: pneumonia, atelectasis, and pulmonary embolisms [30]. In LUS examinations, a “tissue-like pattern” is defined as a homogeneous pattern in a lobe (Figure 5), similar to an abdominal organ parenchyma in the B-mode, and constitutes a real lung anatomical image [20,32]. The “tissue-like pattern” allows one to confirm community-acquired pneumonia diagnoses (sensitivity of 94–99% and specificity of 95–97%) [33,34]. Furthermore, the “shred sign” represents small subpleural consolidation, which is defined by a hypoechoic image delimited by irregular rims in the B-mode. The “shred sign” alone is not highly specific and, conversely, together with a “tissue-like pattern”, supports, with a higher diagnostic accuracy, lung consolidation diagnoses (sensitivity of 90% and specificity of 98%) [35]. Air bronchograms correspond to the air trapped within the lung consolidation region and are defined by linear hyperechoic images within a “tissue-like pattern” [36]. Air bronchograms allow one to distinguish between pneumonia and atelectasis, conditions that share lung consolidation, during LUS examinations [6]. A dynamic air bronchogram, defined by 1 mm of movement during inspiration, corresponds to patent airways and is suggestive of pneumonia with good specificity, ruling out atelectasis (sensitivity of 64% and specificity of 94%) [37]. Atelectasis is a complete loss of aeration within part of the lung or the whole lung. In LUS examinations, atelectasis shows lung consolidation with a “tissue-like pattern” and is associated with the absence of both lung sliding and dynamic air bronchograms [38].

#### 2.3.3. Pulmonary Embolism

For pulmonary embolisms, LUS exams can show the pulmonary infarction area, defined as triangular or rounded, hypoechoic, homogeneous subpleural consolidation [39]. A meta-analysis by Squizzato et al. [40] revealed an overall 87% sensitivity and 82% specificity for pulmonary embolism diagnoses using LUS examinations. According to the BLUE protocol, the combination of an LUS with venous ultrasonography can improve the specificity for pulmonary embolism diagnoses compared to an LUS alone (sensitivity of 81% and specificity of 99%) [1].

## 3. Cardiac Ultrasounds in the Emergency Department

### 3.1. Rationale and Methodology

The principal role of POCUS in the emergency department is the rapid assessment of patients with hemodynamic instabilities [41] through standardized protocols such as the RUSH (rapid ultrasound for shock and hypotension) and the E-FAST protocols (extended focused assessment with sonography in trauma).

Cardiac POCUS is done by using a low-frequency phased array probe (3.5–5 MHz) pointed at four main views: the parasternal long-axis, parasternal short-axis, apical four-chamber, and subcostal/subxiphoid views.

### 3.2. Shock

Shock is a high-mortality syndrome characterized by cellular hypoxia, which can be due to respiratory failure, increased oxygen consumption, or an inadequate supply of O_2_ caused by cardiocirculatory failure and tissue hypoperfusion. This condition is defined by the coexistence of hypotension (systolic BP < 90 mmHg and mean BP < 65 mmHg), tachycardia (HR > 100), and signs of peripheral and central hypoperfusion [42]. There are four main physiopathological settings of shock (Table 1):Distributive.Hypovolemic.Obstructive.Cardiogenic.

POCUS is useful in the early identification of the etiology and of the pathophysiology of shock in the emergency room; in establishing the hemodynamic status through an estimation of the CVP, PWCP, SVR, and CO; in monitoring the patient; and in guiding the therapy [43,44]. In this setting, the RUSH protocol was developed to guide the user in the evaluation of critically ill patients in the stressful emergency setting [45]. This protocol aids the investigation of the most common causes of shock through the evaluation of LV and RV systolic disfunction and cardiac tamponade (Table 2).

**(A)** Estimation of CVP and of blood volume

An estimation of the CVP/RAP (right atrial pressure) can be done through an evaluation of the anteroposterior diameter and grade of collapsibility of the IVC (inferior vena cava) in the subcostal view: a low preload (CVP 0–5 mmHg) is associated with a small diameter of the IVC (<10 mm) and the complete collapse of the IVC with inspiration. A high preload (CVP > 10 mmHg) is characterized by a dilated IVC (>20 mm) and a hypo- (<50%) or noncollapsing IVC [46]. This method is an easy and fast way to assess the patient’s preload and to rule out hypovolemic shock.

**(B)** Fluid responsiveness and stroke volume evaluation

An examination of the IVC alone could fail to answer whether a hemodynamically unstable patient needs to be loaded with fluid [47]. Fluid responsiveness is defined as an increase of 15% in the cardiac output after a rapid infusion of crystalloids (4–7 mL/kg in 15 min) or 1–2 min after a leg-raising test [48]. An echography evaluation of fluid responsiveness necessitates the application of the continuity equation to calculate the stroke volume (SV) variation before and after the volume challenge tests; thus, an evaluation of the LVOT systolic diameter in the parasternal long-axis view would be necessary to estimate the LVOT cross-sectional area (〖CSA〗_LVOT) and the velocity–time integral of the power Doppler of the LVOT that was sampled from the center of the LVOT with a good alignment with the blood flow (normally obtained in the apical five-chamber view):〖CSA〗_LVOT =(π〖LVOT/2)〗^2; SV=〖CSA〗_LVOT〖VTI〗_LVOT; CO = SV × HR

In consideration of the complexity of the evaluation of the LVOT diameter in critically ill patients and of the fact that small measurement errors would be elevated to the second power, we could simplify the equation by comparing the variations of the LVOT VTI or of the sole maximum velocities. This evaluation alone can give the emergency physician extremely important information on the hemodynamics of the patients and on the need for fluid implementation or diuretics administration.

**(C)** Cardiac chambers size and systolic function

An evaluation of the cardiac chamber sizes and biventricular systolic function is of paramount importance in the etiologic diagnosis of hemodynamically unstable patients [49]. In the emergency setting, the LV global ejection fraction is estimated qualitatively as “normal”, “moderately reduced”, or “severely reduced” through an evaluation of the systolic–diastolic fractional shortening of the cavity diameter of the LV, which has shown an adequate correlation with cardiologists’ echocardiographic evaluations [49]. Thus, the identification of reduced LV systolic function can orient the physician towards the diagnosis of cardiogenic shock and indicates a need for further inotropes, for mechanical support, and for an emergent coronary angiographic evaluation [50]. In the evaluation of the RV, it is important to focus on the chamber dimensions (generally 2/3 of the left ventricle size in the apical four-chamber view) and the shape since a D-shaped appearance (Figure 6), associated with septal systolic bulging towards the left ventricle, is suggestive of a high-pressure regimen in the right heart, which could be associated with severe pulmonary hypertension or a pulmonary embolism [51]. RV systolic function can be estimated by evaluating the reduction of the cavity area (normal > 1/3) and the tricuspid lateral annular plane excursion in the M-mode (TAPSE > 15 mm). A dyskinetic RV could suggest the presence of a specific etiology: McConnell’s sign, described as right ventricular free-wall hypokinesis and normal apical contractility, is a specific sign of a pulmonary embolism if associated with RV dilatation and a coherent clinical setting of dyspnea and deep-vein thrombosis (DVT) [52].

**(D)** Diastolic function and Wedge pressure

The diastolic function of the LV can be grossly evaluated in the apical four-chamber view by focusing on the left atrium’s dimensions and through the analysis of the pulsed-wave Doppler (PW) of the transmitral diastolic flow obtained by positioning the volume sample at the tip of the mitral leaflet [53]. The ratio between the protodiastolic E wave and the end-diastolic A wave gives preliminary information on the diastolic function of the heart (fig.). The evaluation of the mitral, annular, septal, and lateral protodiastolic mean velocity through the tissue Doppler technique permits the estimation of the LV diastolic filling pressures by using the E/e’ ratio:-An E/e’ < 8 is generally associated with a normal PWCP.-An E/e’ > 15 is associated with a high PWCP.-An E/e’ between 8 and 15 necessitates a multiparametric evaluation of diastolic function.

This information is useful in guiding the treatment of the patient since, in the setting of diastolic disfunction, with an augmented estimated PWCP, we could expect scarce or no response to fluid administration, while it could be beneficial to unload left heart filling pressures with diuretic therapy. Moreover, the position of the interatrial septum (IAS) can also be helpful: IAS bulging towards the right is a sign of elevated filling pressures in the left cardiac chambers.

**(E)** Pericardium

Emergency-room-focused ultrasounds, using the RUSH protocol or the FAST protocol (described below), permit the detection of pericardial effusion (PE) with an acceptable sensitivity and specificity (Figure 7). The subcostal view can be performed relatively easily in supine patients and is the most appropriate way to diagnose PE, which is visualized as an echo-free space between the heart and the parietal layer of the pericardium. Once a PE diagnosis has been made, the characterization of the fluid and the search for signs of possible cardiac tamponade must be performed by an expert echocardiographer. Pericardial tamponade is a clinical diagnosis that includes hemodynamic instability, pulsus paradoxus, jugular distension [54], and echographic evidence of PE associated with diastolic right chamber collapses and a dilated hypocollapsing IVC.

If an emergency pericardiocentesis is indicated, POCUS can be useful for guiding the insertion of the needle into the pericardial space and for verifying whether the needle is in the pericardial cavity after the infusion of agitated saline solution (Figure 8) [55]. Echo-guided pericardiocenteses have been shown to have an increased possibility of the success of the procedure and a reduced risk of complications when compared to the non-echo-guided procedure [56].

### 3.3. Cardiac Arrest

The patient in cardiac arrest requires a prompt-initiation cardiopulmonary resuscitation (CRP) and Advanced Cardiac Life Support (ACLS) treatment algorithms. In this setting, POCUS could be a useful tool for guiding lifesaving bedside procedures (i.e., pericardiocentesis), for assessing the quality of chest compressions, for confirming the clinical suspect of a reversible cause of CA, and for differentiating true pulseless electrical activity (PEA) from pseudo-PEA [57]. The most recent European resuscitation council guidelines from 2021 [58], which are in line with an ILCOR systematic review [59], recommend the cautious use of POCUS in CA; in particular, no POCUS finding indicative of myocardial infarction, cardiac tamponade, hypovolemia, tension pneumothoraces, or pulmonary embolisms had a sufficiently high sensitivity or specificity to be used as the sole criterion for terminating CPR or for one to “rule out” or “rule in” the cause of cardiac arrest during resuscitation, especially in the absence of the clinical suspicion of a specific reversible cause [60]. For this reason, it is of paramount importance that the image acquisition and evaluation should be carried out by an expert echocardiographer, and it must not interfere with the quality of the resuscitation maneuvers. More specifically the subxiphoid view generally offers good visualization of the IVC, the pericardium, and the heart chambers and valves without impeding chest compressions; thus, it can give immediate information regarding the presence or absence of pericardial effusion, the volume status of the individual, and the quality of the chest compressions, while other views can be utilized in case of a specific clinical suspect and during the reevaluation of the cardiac rhythm. It is important to consider that the dilatation of the RV in the case of cardiac arrest has a low predictive value in diagnosing a pulmonary embolism since it is more probably caused by hypovolemia, hyperkalemia, and ventricular fibrillation, which may occur during cardiac arrest [61]. The possibility of differentiating the electromechanical dissociation of heart activity (PEA) from pseudo-PEA using echography is of particular interest. PEA is characterized by the presence of QRS in an EKG and the absence of cardiac contraction, while pseudo-PEA is defined by the absence of a pulse despite the presence of ventricular contractility being visualized on a cardiac ultrasound. These patients have minimal cardiac output and have a higher survival rate, in part because there are often reversible causes for their arrest that can be identified with POCUS and treated adequately, improving the prognosis and the return to spontaneous circulation [60].

## 4. Point-of-Care Abdominal Ultrasound

The most common POCUS applications of abdominal ultrasounds include the evaluation of patients with abdominal pain or trauma [62,63]. Moreover, POCUS has also been recently used to estimate patients’ fluid status, especially in those with acutely decompensated heart failure and acute kidney injury. A low-frequency (3.5–5 MHz) curvilinear probe is most often used for abdominal USs thanks to its greater depth of penetration, although a linear probe transducer (high-frequency, 5–15 MHz) may be necessary in appendicitis and pediatric applications [64].

### 4.1. Acute Abdominal Pain

Acute abdominal pain represents one of the most common causes of referrals to an emergency department, and POCUS provides help in differentiating patients who require additional diagnostic tests or hospitalization [65,66]. In this setting, US exams can identify intra-abdominal fluids, an aneurysm of the abdominal aorta, and hydronephrosis, and it can also provide important information about patients with trauma [3,67].

**(A)** Intraperitoneal free fluid

Abdominal ultrasounds allow for the detection of pathological intraperitoneal free fluid (IPF) accumulation, although it cannot differentiate the type of liquid. IPF may result from a ruptured ectopic pregnancy, a urine leak, bile, or ascites. IPF appears anechoic and is most often localized in the perihepatic (in Morrison’s pouch) or perisplenic region, while, in the pelvis, it is usually seen in the pouch of Douglas. Therefore, it is important for evaluating the right upper quadrant (RUQ), left upper quadrant (LUQ), periphery of the abdomen (left and right) in the paracolic gutters, and pelvis. Once the fluid is localized, ultrasonographic guidance can increase success and decrease the complications of common procedures such as paracentesis [68]. Most often, IPF may result from thoracic or abdominal trauma. In this setting, focused assessments with sonography (FAST) quickly help to identify the intra-abdominal source of bleeding [3,4]. Indeed, FAST represents a rapid US exam performed by clinicians with the aim to rapidly evaluate patients suspected to have intra-abdominal or intrathoracic free fluid collection or cardiac tamponade. There is also the extended FAST (eFAST) protocol, which includes some additional ultrasound views to identify pneumothoraces [14,15].

**(B)** Appendicitis and Cholecystitis

USs of the right upper quadrant in people suspected to have biliary colic can reveal cholecystitis with a great accuracy [69,70]. Its diagnosis is suggested through the direct visualization of gallstones, which are seen as hyperechoic focuses with posterior shadowing (Figure 9), thickened and stratified gall bladder walls (Fig B), pericholecystic fluid, and a positive sonographic Murphy’s sign [71]. POCUS performed by emergency physician has been shown to be as accurate as radiology ultrasounds in the diagnosis of cholecystitis [72,73], although, sometimes, US interpretation may be challenging owing to the absence of gallstones.

Several studies have demonstrated that the sensitivity and specificity of POCUS in detecting appendicitis are nearly 84% and 90%, respectively [74,75]. When evaluable, the appendix should be examined in the longitudinal and transverse planes [76]. Appendicitis is suspected when a US reveals a dilated noncompressible lumen with a maximum diameter higher than six millimeters, the presence of an appendicolith, and the “Target sign” in the transverse view (Figure 10). However, appendix examinations with USs are affected by the patient body weight, anatomical variants, and operator experience [77].

**(C)** Abdominal aortic aneurysm

POCUS represents an important tool for the diagnosis of abdominal aortic aneurysms in patients admitted into the emergency room for abdominal pain [78]. An AAA is defined as a permanent segmental dilatation of the abdominal aorta which most often involves the infrarenal segment with a maximum diameter over 30 mm. Moreover, large AAAs (over 55 mm) are at a high risk of rupture and should be considered for surgical repair [79]. US examinations must be performed from the epigastrium to the distal bifurcation, and they should describe the aorta’s caliber and shape and differentiate between flap dissection and thrombosis within its lumen [80,81]. Furthermore, an AAA rupture and hemoperitoneum should always be ruled out [3].

### 4.2. Acute Kidney Injury

An acute kidney injury is a common finding among patients admitted to the emergency department, but it portends a bad prognosis and a high mortality rate (about 50%) [82]. Bedside ultrasounds represent an important tool for investigating the etiology of AKIs and for guiding the most appropriate management technique [83].

#### 4.2.1. Hemodynamic AKI

A hemodynamic AKI is due to kidney hypoperfusion, and it may be due to volume depletion; cardiac disfunction; severe abdominal congestion; or vasoplegia, such as in sepsis. A comprehensive assessment of the heart, lung water, and venous congestion [84] is essential to finding the etiology and to guiding therapy (i.e., decongestion, fluid resuscitation, or vasopressor agents) [85] as already discussed [48].

#### 4.2.2. Obstructive AKI

POCUS may quickly identify acute urinary retention and detect obstructive nephropathy requiring procedural drainage [86]. Nephrolithiasis is suggested through the direct visualization of stones [87] or through evidence of indirect signs of obstruction, such as hydronephrosis (Figure 11) or the absence of ureterovesical jets (UVJs). UVJs represent a surrogate for ureteral flow into the bladder, and their absence is most often suggestive of ureterolithiasis [88]. However, obstructive AKIs may be functional or due to intrinsic causes. In this case USs may demonstrate urinary tract dilatation at various levels without the finding of stones.

#### 4.2.3. Renal Resistive Index

Renal vascular studies also give useful information through the assessment of the renal resistive index (RRI). The RRI evaluates the macrovascular perfusion into the kidney [89] and is calculated from the following formula:RRI = [Peak systolic velocity (PSV) − End-diastolic velocity (EDV)/Peak systolic velocity (PSV)],
where the PSV and EDV are obtained at the level of segmental arteries. A greater difference between the peak systolic velocity and end-diastolic velocity leads to a higher RRI and reflects a resistance to the blood flow [90]. The RRI is most often augmented either in intrinsic AKIs or in conditions with an elevated intrabdominal pressure, such as hepatorenal syndrome [91]. Moreover, it has been shown to be predictive of renal outcomes in selected patients [92,93].

#### 4.2.4. Chronic Kidney Disease

Renal ultrasonography provides information on the chronicity of medical renal disease through the evaluation of the renal length, the cortical thickness, and its echogenicity [94]. Hence, the presence of a thinned cortex together with a decreased kidney length suggest chronic kidney disease (CKD). Moreover, in CKD, the cortex appears more echogenic when compared to the liver or spleen even though increased echogenicity is subjective and can be skewed by liver disease [95].

### 4.3. Point-of-Care Venous Doppler Ultrasound

An elevated right atrial pressure portends bad clinical outcomes in selected patients [96]. Venous point-of-care Doppler ultrasonography is emerging as a valuable bedside diagnostic tool for the assessment of fluid overloads, especially in acutely decompensated heart failure [97]. Abnormal flow patterns in hepatic, portal, and intrarenal veins waves have been linked to an elevated RAP, thus providing additive information about end-organ congestion [84]. In brief, congestion indicates poor outcomes in heart failure patients; therefore, a comprehensive assessment of the heart, lung water, and venous congestion permits the real-time evaluation of the efficacy of deresuscitation therapy and bears prognostic significance [84,98].

#### 4.3.1. Inferior Vena Cava

Although the IVC is a good indicator of the central venous pressure (CVP) [99], it is not always reliable for assessing the fluid status, such as assessing that in patients with lung hyperinflation (COPD and asthma), ventilator support, or cardiac conditions impeding the venous return (i.e., RV myocardial infarction and cardiac tamponade) [100]. However, a plethoric IVC (>2.1 cm) with a 50% inspiratory collapse usually indicates a high RAP of 15 mmHg (10–20 mm Hg) [101].

#### 4.3.2. Hepatic Vein Flow

The blood flow patterns in the hepatic veins (HVs) are pulsatile with two retrograde “A” and “V” waves and two anterograde “S” (systolic) and “D” (diastolic) waves. The S wave is larger than the D wave, and they represent the systolic movement of the anulus towards the apex and the diastolic ventricular filling, respectively. When the RAP increases, the amplitude of the S wave decreases when compared to that of the D wave (S < D pattern). As congestion worsens, the S wave becomes blunted until it is reversed [102]. Despite the fluid status, a systolic flow reversal is often observed in severe tricuspid regurgitation, whereas, in advanced pulmonary hypertension, A waves appear elevated together with decreased D wave amplitudes because of the elevated RV end-diastolic pressure [103].

#### 4.3.3. Portal Vein Flow

The portal vein (PV) is isolated from the central veins; therefore, its normal flow pattern appears continuous or mildly pulsatile. The main alteration in the PV waveform is the progressive pulsatility with an increasing RAP, which can be quantified using the pulsatility fraction [104].
PF: [(Vmax − Vmin/Vmax) × 100; a PF ≥ 30% is considered mild, whereas a PF ≥ 50% is considered severe. 

Moreover, an increased pulsatility in the PV waveform is associated with a higher N-terminal pro-brain natriuretic peptide and worse clinical outcomes in heart failure patients if present at discharge [105].

#### 4.3.4. Intrarenal Vein Flow

A normal intrarenal vein Doppler (IRVD) is like that of the portal vein with a continuous flow and a brief interruption during atrial systole [106]. This pattern becomes biphasic with an increased RAP, and two distinct waves (S and D) can be observed, leaving only the diastolic component as congestion worsens (Figure 12). However, an IVRD is, most often, difficult to obtain, and it may be affected by other pathologies, such as severe TR or pulmonary hypertension, despite the volume status. IVRD is also strongly associated with clinical outcomes in HF patients [96].

## 5. Deep-Vein Thrombosis

Acute lower-extremity proximal deep-vein thrombosis (DVT) represents the third leading vascular disease after acute myocardial infarction and stroke [107]. Although phlebography is the gold standard for the diagnosis of DVT [108], ultrasonography has become the standard in clinical practice due to cost effectiveness, quickness, and its simplicity of execution [109,110]. Several studies have shown that POCUS has the same diagnostic power as that of RUSs (radiology department ultrasounds) and venography even when performed in the emergency department [111,112,113].

### 5.1. Protocol

Regarding the instrumentation, the ultrasound machines used range from pocket-sized handset devices to advanced and more sophisticated devices [114]. It is appropriate to perform venous ultrasounds using a linear probe (5–10 MHz) and to set a scan depth from 1 to 4 cm. The position of the patient is certainly important for the correct performance of the examination. The bed must be at the proper height to promote operator comfort. The patient must lie supine with the head slightly elevated while in the “frog leg” position: this consists of extrarotating the thigh and flexing the leg slightly to allow better access to the inguinal region and to the popliteal fossa [114]. The two principal POCUS examination methods studied and applied for the diagnosis of deep-vein thrombosis are the “2-points” and “3-points” exams [115]. Despite the definition, both methods also consider multiple scan points. The diagnosis of deep-vein thrombosis can be performed through the direct visualization of the thrombus. The presence of an anechoic thrombus can be pointed out by performing a “compression ultrasound” (Figure 13): the absence of a reduction in the vessel caliber during compression is a clear sign of DVT [114]. Specifically, the “2-point” POCUS method explores the common femoral vein and popliteal vein [116]. In the “3-point” protocol, the technique is the same as that of the “2-point” examination of the femoral and popliteal veins with the addition of an adjunct projection to scan the proximal region of the femoral vein.

### 5.2. Imaging

The natural history of deep-vein thrombosis goes through several stages: an initial phase (1–6 days) which is characterized by a mobile hypoechoic clot at high risk for embolism; a second stage (within two weeks) that involves a nonhomogeneous structure with mixed echogenicity (Figure 14); and a third and fourth stage (after two weeks) involving thrombus organization and vessel recanalization. Therefore, it is possible to see the flow entering inside of the clot by using the color Doppler technique. The ultrasound stages can be summarized as follows [110]:Acute: the thrombus itself may not be visible. If detected, it is deformable with the force applied on the probe and with a regular surface; it is a distended vein.Subacute thrombus: (before six months and after clot formation) intermediate morphological changes that cannot be included in the chronic phase.Chronic post-thrombotic change: not compressible nor deformable with an irregular surface; the vein caliber may be normal or reduced.

### 5.3. Accuracy of POCUS in Detecting a Pulse during Cardiac Arrest

Assessing the presence or absence of a pulse is a key moment in the management of cardiac arrest. Doppler ultrasounds provide an alternative to the manual pulse check. Sanchez et al. [117] conducted a cross-sectional multipatient, multireader repeated-measures diagnostic study in which they showed that the diagnostic ability to identify carotid pulse restoration during cardiac arrest is far superior in ultrasounds vs. palpation. Similarly, Choen et al. showed how Doppler ultrasounds on the femoral site are more accurate than manual palpation in identifying the presence of a pulse in patients admitted to the ED with cardiac arrest. It has been shown that a cut-off value of the peak systolic velocity of 20 cm/s correlates with a systolic blood pressure value of >60 mmHg [118]. POCUS on the carotid artery has also been evaluated to assess the carotid artery pulsatility during cardiac arrest [119], and it has been shown to rapidly identify the presence of a carotid pulse and, thus, predict ROSC.

### 5.4. TIA/Stroke Clinical Setting

In the clinical setting of suspected stroke or TIA in the emergency department, the POCUS method appears to have a poor correlation with CT for the diagnosis of significant stenosis [120]. Even though Saxhaug et al. showed how the technique is applicable to ruling out significant carotid artery pathology [121], further research and investigation are needed before the extensive use of carotid POCUS in patients with symptoms suggestive of acute cerebral ischemia is carried out.

## 6. Emerging Settings for POCUS Employment in Clinical Practice

### 6.1. Intracranial Pressure Monitoring

POCUS is currently used in various neurological settings, such as intracranial pressure (ICP) estimation, especially after a traumatic brain injury [122]. Even though invasive monitoring devices represent the gold standard, brain ultrasonography (BU) can be used as a valid alternative for intracranial pressure estimation [123]. An increase in the ICP results in the distension of the optic nerve sheath (ONS); therefore, changes in the ONS diameter correlate with ICP variation [124,125]. The ONS diameter (ONSD) can be assessed through the transorbital window using a 2 MHz frequency ultrasound probe. The ONSD is measured behind the globe, using the optic disc as a reference point together with the central retinal artery color Doppler technique, which helps to identify the optic nerve [126]. Moreover, the blood flow velocities in the main cerebral arteries, measured through the transcranial Doppler (TCD) technique, help to identify intracranial hypertension by identifying an elevation in both the pulsatility index (PI) and the resistive index (RI) ([127,128]. In brief, the combination of data such as an optic nerve sheath diameter (ONSD) of >6 mm and a pulsatility index (PI) of >1.4 strongly correlates with an intracranial pressure of >22 mmHg, which identifies intracranial hypertension [129].

### 6.2. Guidance for Invasive Procedures in Critically ill Patients

POCUS also allows one to verify the correct placement of the central venous catheter (CVC) through a saline flush test with evidence of a rapid atrial vortex within 2 s of starting the saline infusion from one of the CVC lumens [130]. Compared to chest radiography, POCUS allows one to verify the CVC position without radiation and with the possibility of identifying complications (e.g., pneumothoraces) in less time and with a noninferior sensitivity [131,132]. Furthermore, in ICUs, upper-airway POCUS provides valid support in the case of difficult intubations, such as those in patients with anatomies distorted by diseases or trauma, where POCUS helps to evaluate the depth of the endotracheal tube or to guide percutaneous tracheostomies [133].

### 6.3. Chest Traumatology

POCUS is also acquiring an emerging role in chest traumatology. In particular, rib fractures are frequent complications of blunt chest trauma [134]. An early diagnosis associated with prompt measures to monitor possible chest trauma complications are essential in the critical care setting [135,136]. POCUS is performed at the site of the patient’s maximal pain. A high-frequency linear probe is positioned perpendicularly to the long axis of the rib. This allows one to identify a hyperechoic structure and its acoustic shadowing, enabling differential diagnoses with adjacent pleural lines [137,138]. Through a US, a rib fracture can be identified as a break in the anterior hyperechoic margin of the rib. The entire rib is explored, and multiple views can be obtained in real time. In the presence of a rib fracture, POCUS can be useful for detecting other associated chest trauma complications, such as PNXs [139] and hemothoraces [140]. Radiography could miss rib fracture diagnoses, especially if these occur in the costal cartilages [141]. In fact, only 49% of rib fractures can be detected by physical evaluations and radiographs [142]. Conversely, POCUS had a sensitivity of 91.2% and a specificity of 72.7% in diagnosing rib fractures [143]. However, pain related to probe pressure [144] and inaccessible but uncommon fracture sites, such as the retroscapular ribs and the infraclavicular segment of the first rib, could represent major limitations to POCUS examinations [145].

### 6.4. The emerging Role of Artificial Intelligence in POCUS

The implementation of artificial intelligence (AI) software in POCUS is undoubtedly fascinating and carries a great potential improvement to this methodology. The field of the application of machine learning and of deep learning software in echography could be directed at the training of resident physician [146]; nonetheless, it can support expert physicians in the emergency setting in reducing the duration of the exam via automated measurements with a reduction in the interoperator variability [147]. Several FDA approved machine learning softwares are already implemented in modern echography devices as reported by Knackstedt et al. [148], and the automeasurement of the left ventricular ejection fraction, of the left ventricle end-systolic and end-diastolic volumes, and of the global longitudinal strain showed a consistent agreement with manual measurements. Similarly, an auto-LVOT-VTI calculating software presented a good correlation with thermodilution cardiac output measurements [149]. Moreover, Zhang et al. [150] found that a deep learning software using convolutional neural networks (CNNs) had a 96% accuracy in discriminating the echocardiographic view and a high accuracy in diagnosing several cardiac structural abnormalities. Artificial intelligence may hasten lung ultrasound evaluations through the automatic calculation of the Lung Ultrasound Score (LUS) and by helping in the discrimination of A-lines, B-lines, consolidation, and pleural effusion, with a potential utility in the emergency setting and in COVID-19 patients [151,152,153]. In the present day, AI can represent a promising tool for the improvement of POCUS evaluations, but it still needs further development and the implementation of large datasets to become an ally of the physician.

## 7. Conclusions

POCUS represents a goal-directed ultrasound examination which allows clinicians to rapidly evaluate patients in the emergency setting through a limited and focused number of standard echographic views. POCUS helps to make quick diagnoses and to identify early life-threatening conditions which require prompt interventions. The aim of our review was to summarize the various POCUS applications with a focus on the emergency setting as discussed below. Indeed, even if initially meant for cardiac and lung disease evaluations, US exams have been extended to many medical conditions with great clinical benefits both in the emergency setting and in critically ill patients. These considerations make ultrasonography an integral part of emergency medicine.

## Figures and Tables

**Figure 1 ijerph-20-02088-f001:**
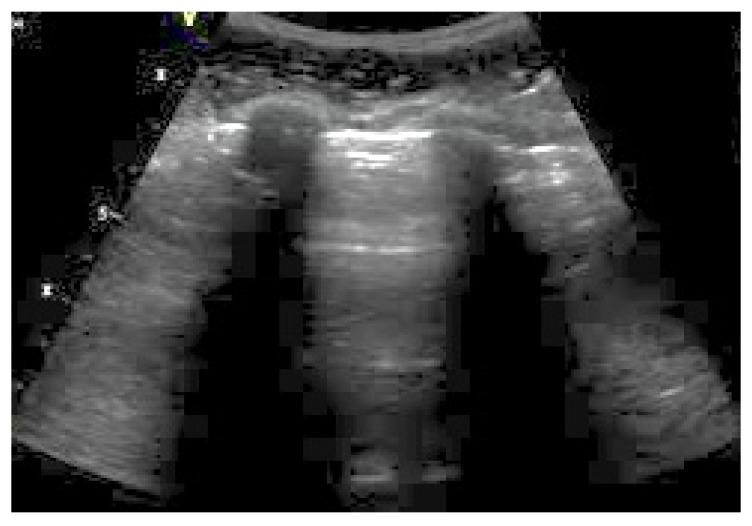
A-lines (arrow).

**Figure 2 ijerph-20-02088-f002:**
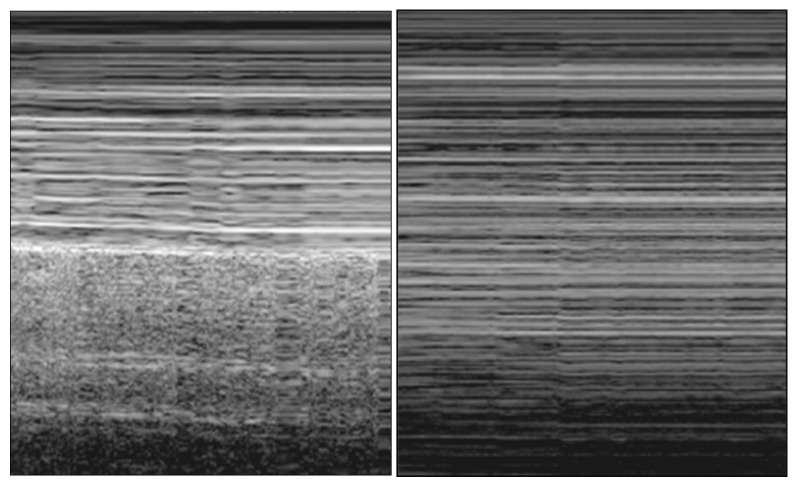
Lung point: “seashore sign” (**left**) turns into “stratosphere sign” (**right**) in M-mode view during deep inspiration.

**Figure 3 ijerph-20-02088-f003:**
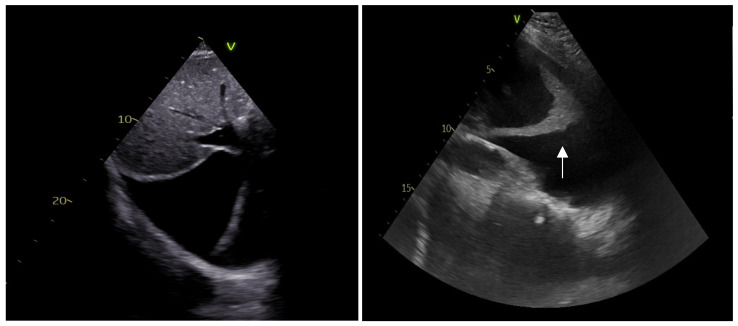
Pleural effusion with lung region floating inside (arrow).

**Figure 4 ijerph-20-02088-f004:**
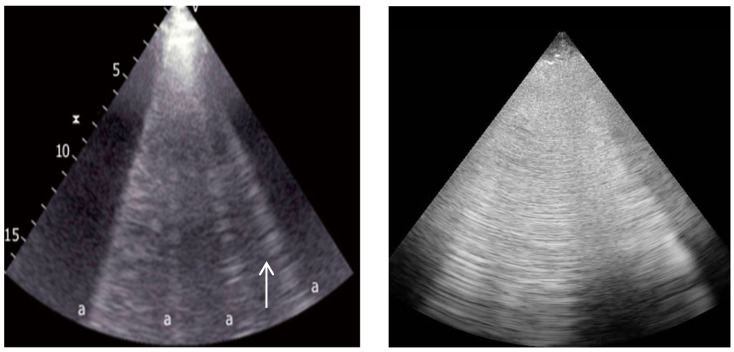
B-lines (**left**, indicated by arrow) and “B-pattern” (**right**).

**Figure 5 ijerph-20-02088-f005:**
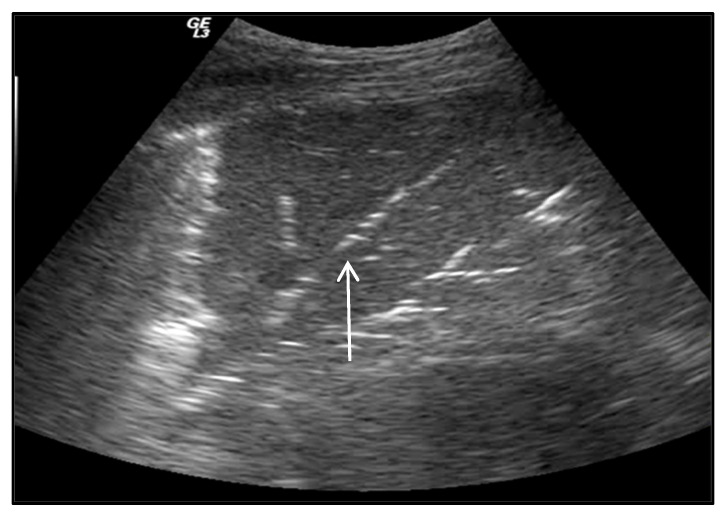
“Tissue-like pattern” with air bronchogram inside (arrow).

**Figure 6 ijerph-20-02088-f006:**
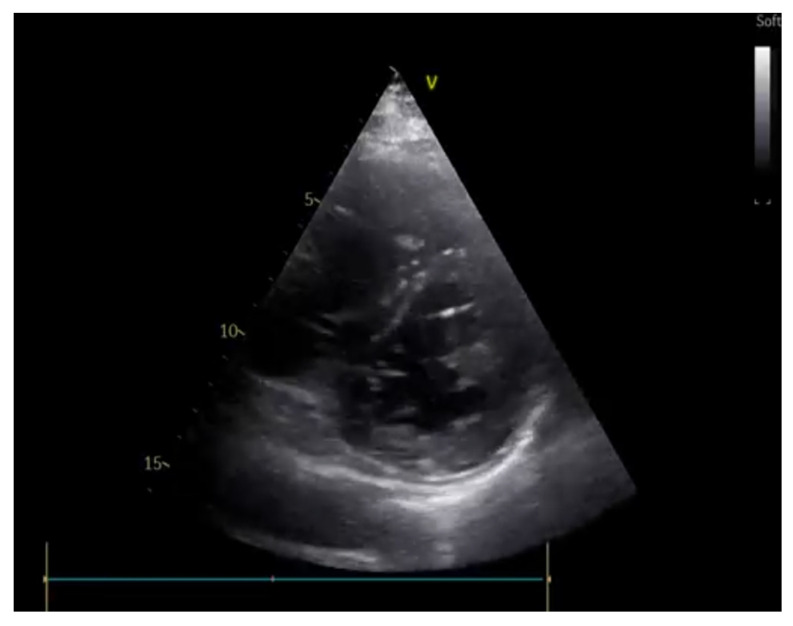
Systolic bowing of interventricular septum towards the left ventricle (D-shape) is associated with pulmonary embolism and pulmonary hypertension.

**Figure 7 ijerph-20-02088-f007:**
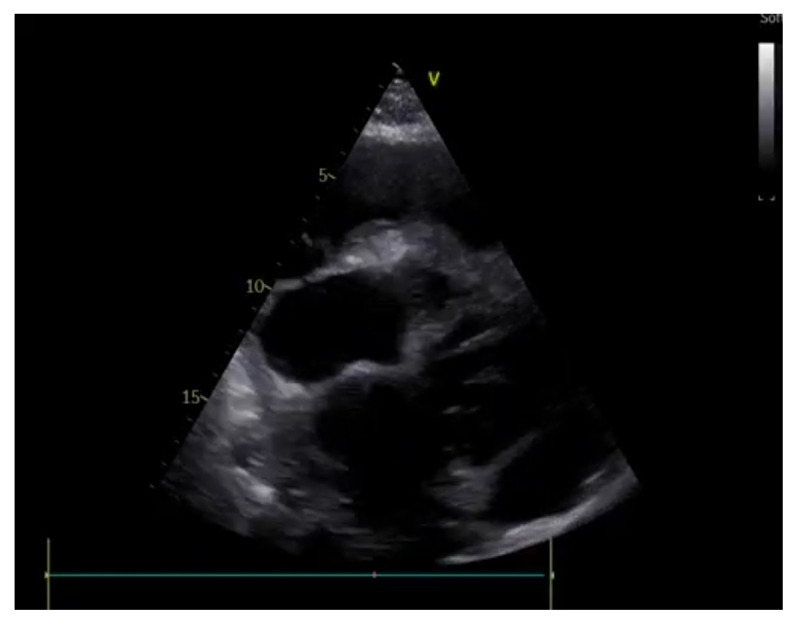
Pericardial tamponade: end-diastolic right ventricular chamber collapse is associated with significant pericardial effusion.

**Figure 8 ijerph-20-02088-f008:**
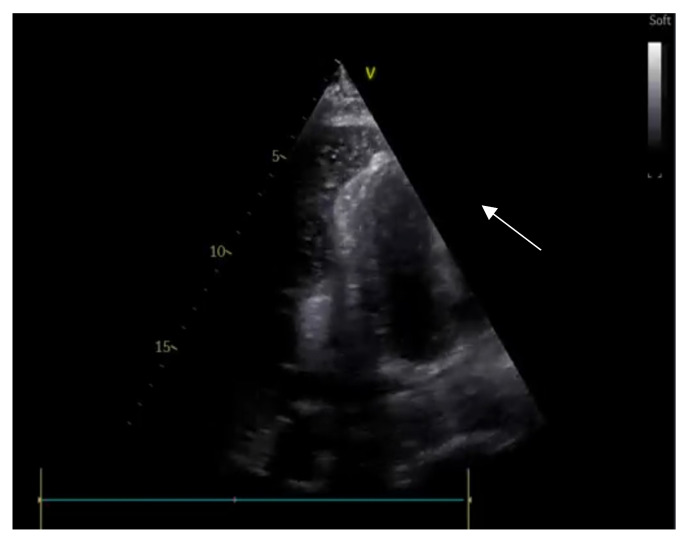
Bubble test during pericardiocentesis: appearance of bubbles in the pericardium (arrow) indicates that the needle is correctly positioned in the pericardial space.

**Figure 9 ijerph-20-02088-f009:**
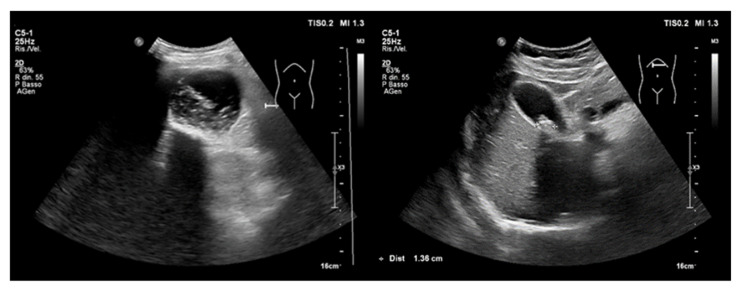
Ultrasonography of the right upper quadrant. A distended gallbladder with microlithiasis and large biliary sludge (**left**); multiple stones with posterior shadowing in the infundibulum (**right**).

**Figure 10 ijerph-20-02088-f010:**
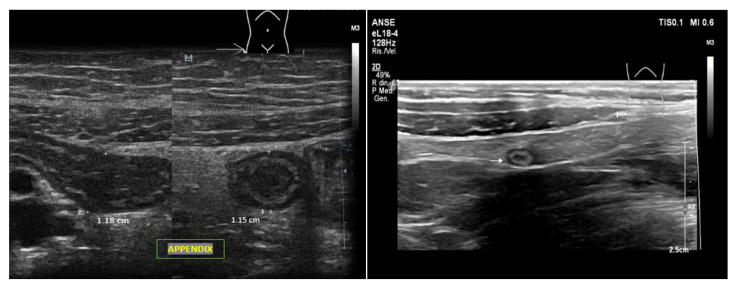
Ultrasonography of normal and abnormal appendixes. On the left, there are longitudinal and transverse USs of acute appendicitis showing thickened walls and diameter > 6 mm. On the right, there is a thinner normal appendix.

**Figure 11 ijerph-20-02088-f011:**
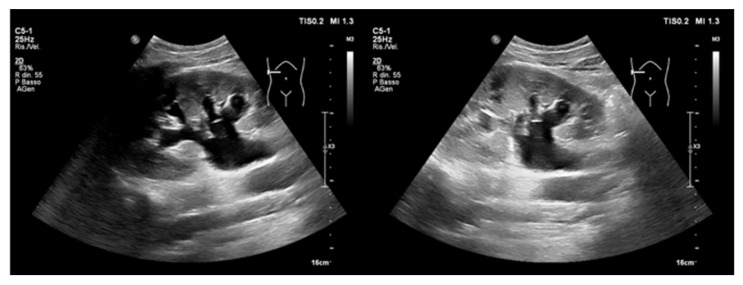
Pelvis and caliceal dilatation with proximal ureteral stone.

**Figure 12 ijerph-20-02088-f012:**
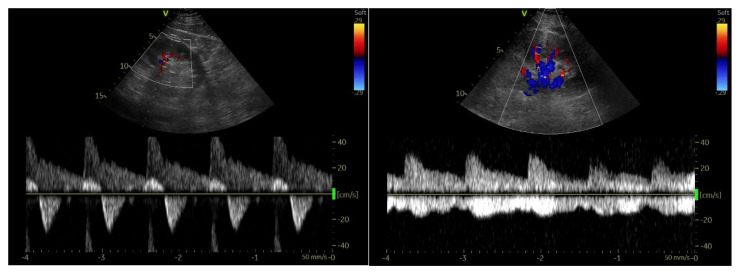
Intrarenal vein Doppler (above baseline) in a patient with acutely decompensated heart failure before (**left**) and after (**right**) decongestive therapy. The flow pattern appears monophasic/diastolic (L), becoming continuous as congestion reduces (R).

**Figure 13 ijerph-20-02088-f013:**
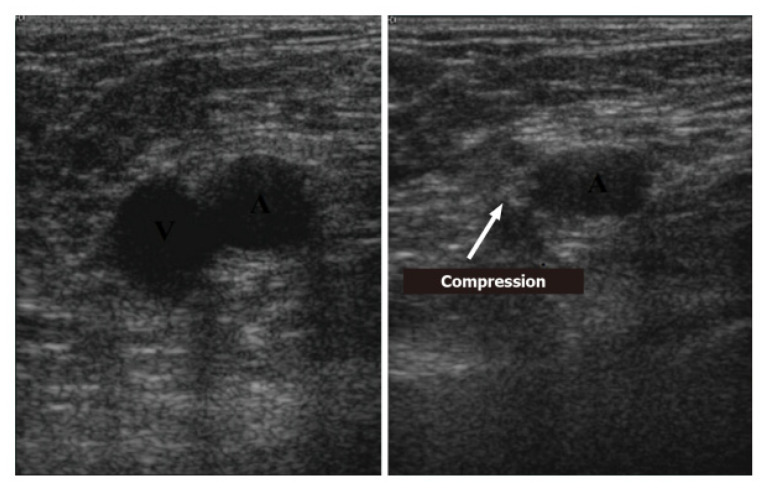
Compression ultrasound [Di Vilio A, Vergara A, Desiderio A, et al.: “A. Incremental value of compression ultrasound sonography in the emergency department. World J Crit Care Med. 2021 Sep 9;10(5):194–203].

**Figure 14 ijerph-20-02088-f014:**
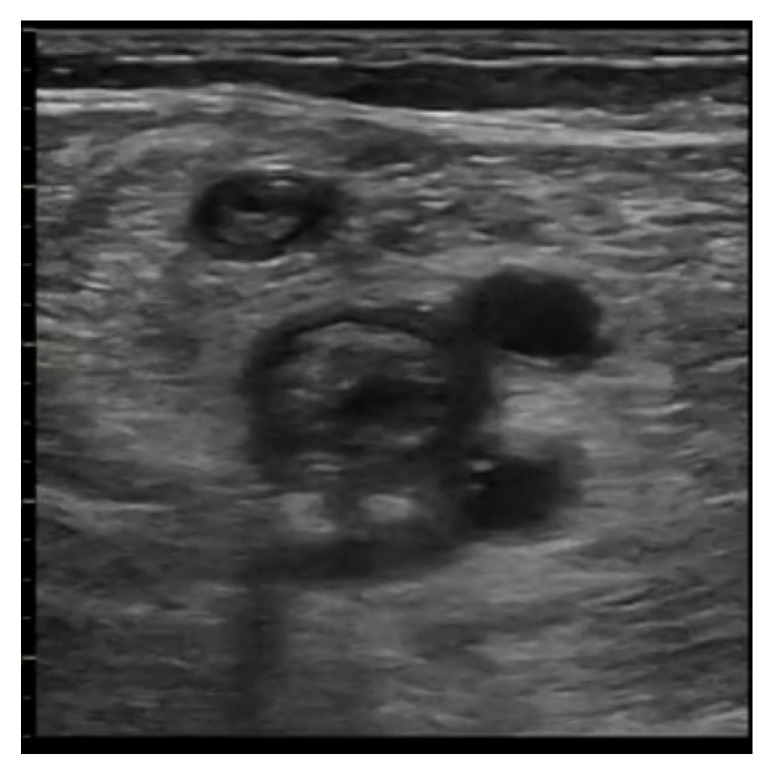
The image shows the presence of an extensive thrombus with mixed echogenicity occupying the lumen of the femoral vein and the great saphenous vein.

**Table 1 ijerph-20-02088-t001:** Types of Shock [Mcgee W., Adler A., Sharma R., et al. Hemodynamic Assessment and Monitoring in the Intensive Care Unit: an Overview. Journal of Anesthesiology and Critical Care 01 Aug 2014. Doi:10.18650/2374-4448.14010].

Types of Shock		
Type	Etiology	Hemodynamics
Distributive	Septic, neurogenic, anaphylactic.	↓ SVR ↓CVP ↓ PWCP ↑ CO (Hyperdynamic state) ↓ CO (Hypodynamic state)
Hypovolemic	Hemorrhagic, dehydration, third space sequestration.	↑ SVR ↓ CVP ↓ PWCP ↓ CO
Cardiogenic	Myocardial infarction, advanced heart failure, brady/tachyarrhythmias, valvular disease.	↑ SVR ↑ CVP ↑ PCWP (LV disfunction) ↓ PCWP (RV disfunction) ↓ CO
Obstructive	Heart tamponade, pulmonary embolism, tensive pneumothorax, constrictive pericarditis, restrictive cardiomyopathy.	↑ SVR ↓ CVP ↓ PCWP (RV disfunction) ↑ PCWP ↓ CO

Abbreviations: CO represents cardiac output, CVP represents central venous pressure, PVR represents peripheral vascular resistance, and PWCP represents pulmonary capillary wedge pressure.

**Table 2 ijerph-20-02088-t002:** RUSH protocol. Echography finding in different types of shock [45].

RUSH Protocol				
	Hypovolemic	Distributive	Cardiogenic	Obstructive
**Pump**	- Hypercontractility. - Small dimensions of LV. - “Kissing” LV walls in systole.	- Hyper- or hypocontractility.	- Hypocontractility. - Normal dimensions or dilation of LV.	- Dyskinetic RV (McConnell’s sign). - Intraventricular thrombus. - Cardiac tamponade.
**Tank**	- Small or virtual IVC. - Look for pleural and peritoneal fluid.	- Normal or small IVC. - Look for causes of sepsis.	- Dilated and hypocollapsing IVC. - Look for ascites, pleural effusion, and pulmonary congestion.	- Dilated and hypocollapsing IVC. - Look for PNX (Lung point).
**Pipes**	- Aortic aneurism. - Aortic dissection.	- Normal.	- Normal.	- DVT.

Abbreviations: DVT represents deep-vein thrombosis, IVC represents inferior vena cava, LV represents left ventricle, and PNX represents pneumothorax.

## Data Availability

Not applicable.

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
