# Peer review of "The Incremental Role of Multiorgan Point-of-Care Ultrasounds in the Emergency Setting"

_ijerph, 2023, doi:10.3390/ijerph20032088_

Round 1

Reviewer 1 Report

This paper summarized some applications of POC ultrasound in the emergency setting, including echocardiography, lung ultrasound, screening for deep vein thrombosis in people suspected of pulmonary embolism, abdominal ultrasonography. The work is meaningful for the community since POC ultrasound is showing increasing applications to different scenarios. 

However, not going to details of the paper, obviously the content of the article was not comprehensive, although the authors analyzed relevant work covering lung, cardiac, abdomen and vein. As a review paper, it should include the efforts in head (e.g., monitoring intracranial pressure monitoring), neck (e.g., guidance of trachea cannula), as well as internal bleeding, which are important emergent trauma. With such a concern, the authors are encouraged to cover more clinical applications. 

In addition, artificial intelligence involved in those areas as emerging techniques should also be discussed. 

English writing should be improved by native speakers.

Author Response

Dear reviewer,

thanks for the suggestions.

We have modified our original paper accordingly as shown by the attachment.

We  included the following sections:

  • Intracranial pressure monitoring
  • guidance of tracheostomy
  • Internal bleeding 
  • the role of artificial intelligence in the emergency setting

Best regards

Reviewer 2 Report

Dear Authors and Editors, 

I have read the article entitled "The incremental role of the multi-organ POCUS in the energency setting with great interest.

The tittle describe the core message of the paper.

The abstract incorporates key message, in a concise manner.

The structure of the paper is accurate.

Hovewer, I have some suggestions regarding this paper.

1. Please add some informations about place of POCUS in actuall guidelines, this will add addittional value for the paper.

2. Please precise, why this topic is relevant, and what additional value your paper gives, when compared it to other published research. 

3. Please add some informations about assesment of fractured ribs, and about usefulness of US in quick assesment of fractured CVC vs retained calcified fibrin sheath. 

Author Response

Dear reviewer,

thanks for the suggestions.

We have modified our original paper accordingly as shown by the attachment.

We  included the following sections:

  • emerging POCUS applications
  • Role of POCUS in trauma setting (like ribs fracture)
  • US exam for invasive procedure guidance

Best regards

Round 2

Reviewer 1 Report

The revised version has addressed the reviewer’s suggestions. One needs to address is that the suggested fields were not mentioned in the abstract.

Reviewer 2 Report

The paper was significatly improved. In present form it has great value and summarise current knowledge about POCUS in a concise manner.